# Identification of NS2B-NS3 Protease Inhibitors for Therapeutic Application in ZIKV Infection: A Pharmacophore-Based High-Throughput Virtual Screening and MD Simulations Approaches

**DOI:** 10.3390/vaccines11010131

**Published:** 2023-01-05

**Authors:** Hafiz Muzzammel Rehman, Muhammad Sajjad, Muhammad Akhtar Ali, Roquyya Gul, Muhammad Irfan, Muhammad Naveed, Munir Ahmad Bhinder, Muhammad Usman Ghani, Nadia Hussain, Amira S. A. Said, Amal H. I. Al Haddad, Mahjabeen Saleem

**Affiliations:** 1School of Biochemistry and Biotechnology, University of the Punjab, Lahore 54590, Punjab, Pakistan; 2Department of Human Genetics and Molecular Biology, University of Health Sciences, Lahore 54590, Punjab, Pakistan; 3School of Biological Sciences, University of the Punjab, Quaid e Azam Campus, Lahore 54590, Punjab, Pakistan; 4Faculty of Life Sciences, Gulab Devi Educational Complex, Lahore 54590, Punjab, Pakistan; 5Kauser Abdulla Malik School of Life Sciences, Forman Christian College (A Chartered University), Lahore 54600, Punjab, Pakistan; 6Department of Biotechnology, Faculty of Science and Technology, University of Central Punjab Lahore, Lahore 54590, Punjab, Pakistan; 7Center for Applied Molecular Biology, University of the Punjab, Lahore 54590, Punjab, Pakistan; 8Department of Pharmaceutical Sciences, College of Pharmacy, Al Ain University, Al Ain 64141, United Arab Emirates; 9AAU Health and Biomedical Research Center, Al Ain University, Abu Dhabi 112612, United Arab Emirates; 10Department of Clinical Pharmacy, College of Pharmacy, Al Ain University, Al Ain 64141, United Arab Emirates; 11Clinical Pharmacy Department, Faculty of Pharmacy, Beni Suef University, Beni Suef 62521, Egypt; 12Chief Operations Office, Sheikh Shakhbout Medical City (SSMC) in Partnership with Mayo Clinic, Abu Dhabi 11001, United Arab Emirates; 13School of Medical Lab Technology, Minhaj University Lahore, Lahore 54770, Punjab, Pakistan

**Keywords:** Zika virus, e-pharmacophore approach, high-throughput virtual screening, ASINEX database, prime MM-GBSA, molecular dynamics simulation

## Abstract

Zika virus (ZIKV) pandemic and its implication in congenital malformations and severe neurological disorders had created serious threats to global health. ZIKV is a mosquito-borne flavivirus which spread rapidly and infect a large number of people in a shorter time-span. Due to the lack of effective therapeutics, this had become paramount urgency to discover effective drug molecules to encounter the viral infection. Various anti-ZIKV drug discovery efforts during the past several years had been unsuccessful to develop an effective cure. The NS2B-NS3 protein was reported as an attractive therapeutic target for inhibiting viral proliferation, due to its central role in viral replication and maturation of non-structural viral proteins. Therefore, the current in silico drug exploration aimed to identify the novel inhibitors of Zika NS2B-NS3 protease by implementing an e-pharmacophore-based high-throughput virtual screening. A 3D e-pharmacophore model was generated based on the five-featured (ADPRR) pharmacophore hypothesis. Subsequently, the predicted model is further subjected to the high-throughput virtual screening to reveal top hit molecules from the various small molecule databases. Initial hits were examined in terms of binding free energies and ADME properties to identify the candidate hit exhibiting a favourable pharmacokinetic profile. Eventually, molecular dynamic (MD) simulations studies were conducted to evaluate the binding stability of the hit molecule inside the receptor cavity. The findings of the in silico analysis manifested affirmative evidence for three hit molecules with −64.28, −55.15 and −50.16 kcal/mol binding free energies, as potent inhibitors of Zika NS2B-NS3 protease. Hence, these molecules holds the promising potential to serve as a prospective candidates to design effective drugs against ZIKV and related viral infections.

## 1. Introduction

A mosquito-borne pathogen, Zika virus (ZIKV) is an RNA virus from the Flaviviridae family and exhibits significant association with congenital malformations as well as severe neurological disorders [1]. The ZIKV is mainly transmitted by Aedes (subgenus *Stegomyia*) mosquito [2]. It emerged in 2007 in Yap Island in the western Pacific [3] and re-emergened in 2013–2014 in French Polynesia and the South Pacific [4]. For the first time, serious consequences and non-vector-borne transmission of ZIKV were documented in this pandemic. A national public health emergency was declared in 2015 after the Zika virus outbreak in Brazil, and a worldwide public health emergency in was declared in 2016 after reports of rising microcephaly cases. Later on, the Zika virus spread out to the continental USA, Africa, and Southeast Asia in 2016–2017. It was confirmed that ZIKV also follows non-vector-borne transmission and was also implicated in severe neurological problems in adults, neonates and in fetuses [5].

ZIKV is a spherical and single stranded virus containing 11 kilobases RNA genome [6]. The open reading frame (ORF) encodes a polypeptide having 3423 amino acids. This polypeptide is cleaved by the host, as well as viral proteases, into a total of 10 individual proteins, of which 3 are structural and are 7 non-structural proteins. The structural proteins are capsid (C), premembrane (prM), and envelope (E) proteins, whereas non-structural (NS) includes NS1, NS2A, NS2B, NS3, NS4A, NS4B, and NS5 [7]. Among all the non-structural proteins, the NS2B-NS3 protein plays an imperative role in viral replication and maturation of non-structural viral proteins. Hence, potent inhibitors of NS2B-NS3 protease could translate into a promising treatment for ZIKV infections [8]. It exhibits similarities with other *flaviviruses* such as tick-borne encephalitis virus, Japanese encephalitis virus, yellow fever virus, and West Nile virus, but possesses close similarities mainly with the dengue virus (DENV), exhibiting immunological cross-reactivity [9]. The NS2B-NS3 protein also plays a pivotal role in the downregulation of the DENV-triggered antiviral responses, known to be executed by immune evasion of the NS2B-NS3 protein [10]. Hence, it primarily serves as a dual function target, and its inhibition can hinder viral replication with the subsequent protection of innate immunity.

Notwithstanding the severity of the epidemic and several drug discovery efforts during the past years, no treatment had been developed. Efficacious drug discoveries necessitate intricated, expensive, and time-intensive processes. The e-pharmacophore-based high-throughput virtual screening holds the promising potential to identify clinically effective drugs at the cost of minimum time and expense [11,12,13]. The afore mentioned approach has precedence over other drug repurposing strategies as reported for several discoveries of specific inhibitors for the potential control of viral infections [14,15]. For this purpose, the present study was designed to generate energy-optimised pharmacophore by employing the Pharmacophore Alignment and Scoring Engine (PHASE) module of Schrödinger Suite to achieve the successful screening of the molecules having the potential to inhibit NS2B-NS3 protein of Zika virus.

## 2. Materials and Methods

### 2.1. Molecular Databases Used

The co-crystallised structure of the NS2B-NS3 and boronate (PDB ID: 5LC0) was retrieved from the Protein Data Bank (PDB), having a resolution of 2.7 Å [16]. The PubChem database was explored to retrieve the 2D structure of the boronate inhibitor (PubChem ID: 16740933). It has inhibitory activity for ZIKV NS2B-NS3, with half maximal inhibitory concentration (IC_50_) = 0.25 ± 0.02 μM and inhibition constant (*K*_i_) = 0.040 ± 0.006 μM [17]. The ASINEX database, which contains a regularly updated collection of compounds with several libraries covering distinct chemical characteristics, was used for the virtual screening of the compounds to reveal novel hits [18].

### 2.2. Ligand Preparation

The LigPrep module of the Schrödinger package was employed to prepare the ligand [19] by the modification of the ligand’s torsions, followed by the assigning of protonation states [20]. Subsequent conversion of the 2D ligand structure into 3D, hydrogen attachment, generation of stereoisomers, and ionization state identification lead to ligand refining [21,22]. Furthermore, the OPLS4 force field incorporated in PHASE was used for energy minimization and to optimize the low-energy 3D structure of the ligand.

### 2.3. Protein Preparation

The retrieved crystal structure of the NS2B-NS3 protein (PDB ID: 5LC0) was refined using Schrödinger’s protein preparation wizard [23]. This ensured high-confidence structural correctness by transforming the protein structure from the raw state to the refined state and to be prepared adequately for molecular docking and molecular dynamic studies. The protein preparation process involved assigning the corrected bond orders, omitting water molecules and other non-specific chemical components from the crystal structure, and the addition of hydrogen atoms to the protein structure for remodelling the tautomeric and ionization states of amino acid residues [24]. Subsequent addition of missing hydrogen and loops in the structure was followed by restrained energy minimization by employing the OPLS4 force field to achieve high accuracy [25].

### 2.4. Molecular Docking

Schrödinger’s GLIDE module was employed to perform docking of the NS2B-NS3 protein together with the prepared ligand. This facilitated the enhancement of the binding affinity by contributing to the identification of specific structural motifs [26]. XP descriptors were used to analyze the docking results, which provided significant details of various intermolecular interactions [27]. The energy for each pose was identified by Glide XP descriptor information, to optimise the ligand conformation of the ligand–receptor complex. Prior to performing the docking protocol, Schrödinger’s Glide Grid Generation was utilised to generate a Glide Grid around the co-crystallised boronate ligand [28]. Grid generation was followed by the docking process using the prepared ligand and protein structures. Subsequently, the resultant protein–ligand complex of NS2B-NS3 and boronate was used to generate e-pharmacophore model.

### 2.5. Energy-Optimised Pharmacophore

The intrinsic flexibility of a target active site was incorporated into energy-optimised pharmacophore (e-pharmacophore) model generation. According to reports, energetically efficient pharmacophores can screen millions of molecules in a short amount of time. The e-pharmacophore methods combine the features of ligands with structure-based approaches to study the intrinsic flexibility of active sites and ligand binding [29]. An e-pharmacophore model was developed by using the protein–ligand complex of NS2B-NS3 and boronate in the phase module of Schrödinger [30]. The default six pharmacophore features include the aromatic ring (R), the negatively ionizable region (N), the hydrogen bond acceptor (A), the hydrogen bond donor (D), the hydrophobic group (H) and the positively ionizable region (P) were employed in the Phase module [31]. The particular amount of energy comparable to the summation of Glide XP contributions was assigned to each pharmacophore feature. This aided in quantifying and ranking the sites concerning the energy values. The e-pharmacophore hypotheses were generated based on molecular patterns known as SMARTS patterns (Smiles ARbitrary Target Specification patterns) [32].

### 2.6. ADME/T Analysis and High-Throughput Virtual Screening

A set of molecules was generated by virtual screening on the basis of the properties of the e-pharmacophore. These compounds were subjected to molecular docking with NS2B-NS3 as selected molecules possessing common descriptors of e-pharmacophore [28]. Absorption, distribution, metabolism, excretion, and toxicity (ADMET) analysis was conducted to determine numerous properties of these docked molecules, which are required to evaluate their potential as prospective inhibitors of NS2B-NS3. Absorption, distribution, metabolism, excretion, and toxicity (ADMET) are important factors in the development of new drugs. A high-quality drug candidate should exhibit the right ADMET characteristics at a therapeutic dose in addition to having enough activity against the therapeutic target. The two main factors that contribute to medication failure are ineffectiveness and safety; therefore, chemical properties, including absorption, distribution, metabolism, excretion, and toxicity (ADMET), are crucial at every stage of drug discovery and development. Finding effective compounds with improved ADMET characteristics is thus important. Hence, Lipinski rules for the top hits were also assessed, and the properties assessed included molecular weight, brain/blood partition coefficient, lipophilicity (log Po/w), hydrophobicity, hydrophilicity (QPlogPo/w, QPlogS, QPPCaco, QPlogBB, and QPPMDCK), and human oral absorption (HOA%) [33]. Molecular docking was employed to filter out the binding compounds in order to obtain the finest corresponding binding pose of ligands to the protein. It aimed to generate several poses (possible conformations) of the inhibitor within the binding site of the protein. Hence, protein–ligand complexes were generated and filtered based on scoring, representing their binding affinity [34,35]. Moreover, virtual screening was performed by using e-pharmacophore model coupled with high-throughput virtual screening (HTVS), standard precision (SP), and extra precision (XP) docking to achieve a set of molecules with promising binding energies [36].

### 2.7. Binding Free Energy Calculation Using MM-GBSA

To quantify the binding affinity between the protein and small molecules, the molecular mechanics-generalized born surface area (MM-GBSA) approach was adopted. The prime module of Schrödinger was used to estimate the theoretical binding free energies of the promising NS2B-NS3 binding small molecules. The hits obtained after the XP docking, based on the Glide scores, were subjected to MM-GBSA analysis [37]. The energetics of free ligand, free protein, and protein–ligand complex were used to estimate the binding free energies [38].

ΔG_bind_ calculations were executed using the following equations:ΔG_bind_ = G_com_ − (G_rec_ + G_lig_)(1)

Using Equation (1), ΔG_bind_ is obtained by the difference between total of the free energies of the ligand (G_lig_) and the receptors (G_rec_) and the free energy of protein–ligand complex (G_com_).

ΔG_bind_ is supposed to comprise enthalpy (H) [38] (Onufriev et al., 2004) and entropy (TΔS) as presented in Equation (2).
ΔG_bind_ = ΔH − TΔS(2)

### 2.8. Molecular Dynamics Simulation 

The Desmond module of Schrodinger was exploited to conduct MD simulation studies. The dynamic behavior and stability of the protein–ligand complexes were investigated using their docked poses and the co-crystalized structure of NS2B-NS3, and boronate was used as a control [39]. The protein–ligand complexes were preprocessed using Protein Preparation Wizard of Maestro, which included complex optimization and minimization. All the systems were prepared using the System Builder tool. Solvation of the complexes was performed with the simple point-charge (SPC) water model with orthorhombic box, along with a 10-Å distance from the edge of the box, and the system was neutralized with Na^+^/Cl^−^ ions. To mimic physiological conditions, 0.15 M sodium chloride (NaCl) was added. The potential energy of the protein–ligand complexes was minimized by employing the NPT ensemble. The molecular simulations were performed at 300 K temperature and 1 atm pressure for 100 ns and NPT production ran under the OPLS4 force field. The models were relaxed before the simulation. The trajectories were saved for examination after every 100 ps, and the simulation’s stability was verified by comparing the protein and ligand’s Root Mean Square Deviation (RMSD) over time. The projected changes in their conformation from the initial structure over the entire simulation period were expressed as Root Mean Square Deviation (RMSD) and Root Mean Square Fluctuation (RMSF) for MD simulations.

## 3. Results

### 3.1. E-Pharmacophore Model

As a qualitative process, energetically optimized pharmacophore modelling, already assist to determine novel moieties with several biological activities [40]. The present study aimed to predict the e-pharmacophore model to screen NS2B-NS3 inhibitors, using Schrödinger’s PHASE module. To serve the purpose, the structure and energy-based e-pharmacophore generation approaches were employed using the obtained values in the scoring function of GLIDE [41]. The pharmacophore hypothesis was generated using the PHASE module’s six intrinsic pharmacophore features. The pharmacophore hypothesis exhibited five pharmacophoric features (ADPRR) including one hydrogen bond acceptor (A), one hydrogen bond donor (D), one positively ionic (P) and two aromatic rings structures (R) (Figure 1). The higher correlation indicates greater efficiency of the predicted model to assess the activities of potential inhibitor molecules [42].

### 3.2. E-Pharmacophore-Based High Throughput Virtual Screening

The potential NS2B-NS3 protein inhibitors were retrieved from the ASINEX database that contains screening libraries of lead-like molecules. The generated pharmacophore model (ADPRR) was used to screen the database, and the molecules were retrieved from the database by using formulated features of the pharmacophore hypothesis. To find out the potential inhibitor of the NS2B-NS3 protein, these molecules were subjected to docking-based virtual screening consisting of HTVS, SP, and XP docking steps, followed by the validation of hits through MM-GBSA calculations. For molecular docking, the GLIDE module of Schrödinger was employed [43,44]. Initially, the binding efficiency of molecules was estimated by performing HTVS of 25% hit molecules against the binding site of the NS2B-NS3 protein. The, docking accuracy was increased by conducting 25% SP docking, using top compounds, which eliminated false positives to enhance the accuracy. The 25% filtered molecules were then used for XP docking, which resulted in 285 hits. The hits were found to possess scores ranging from −8.838 to −5.676 kcal/mol. The glide scores and MMGBSA values of the top three hits are summarized in Table 1.

### 3.3. Description of the ADME Properties of the Hit Molecules

Absorption, distribution, metabolism, and excretion (ADME) properties in acceptable ranges are the prime requisites for drug designing. Therefore, the pharmacokinetic profile of the hits is mandatory to prevent any failure during the drug development process [45]. Pharmacokinetic profiles of hits and the findings of the ADME investigations are shown in Table 2. The most significant descriptors taken into consideration include QPlogPo/w, QPlogS, QPlogBB, and QPPMDCK (determining hydrophobicity and hydrophilicity), #star (showing number of properties that fall beyond the recommended range), and Rule of Five (indicating number of violations of Lipinski’s rule of five). The findings showed all top three hit molecules possess acceptable ADME properties, with the potential hits containing promising pharmacokinetic properties.

### 3.4. Intermolecular Interactions of Hit Molecules with NS2B-NS3 Protein

Intermolecular interaction patterns of the NS2B-NS3 protein structure were determined using Schrödinger’s ligand interaction diagram that displays hydrogen bonds and salt bridges as intermolecular interactions [46]. The hydrogen bonds play a pivotal role in stabilizing the protein–ligand complex [47]. For NS2B-NS3 protein, the amino acid residues that exhibited hydrogen bonding with compound 1 include Ser 1135, Gly 1151, Gly 1153, and Tyr 1161. Compound 1 also made a salt bridge with Ser 1135, as indicated in Figure 2. Compound 2 made four hydrogen bonds with Tyr 1130, Gly 1151, Asp 1075, and Asn 1152, and pi–pi stacking with Tyr 1161 (Figure 3). Compound 3 formed two pi–pi stackings with His 1051 and Tyr 1150, and six hydrogen bonds with Asp 83, Phe 84, Asn 1152, Ser 1135, Tyr 1161, and Tyr 1130 (Figure 4).

### 3.5. Binding Free Energy Calculation Using MM-GBSA

The MM-GBSA approach was adopted to calculate the binding free energies of Comp1, Comp2, and Comp3 to assess their binding affinity with the NS2B-NS3 protein. The calculated ΔG_bind_ for these compounds is given in Table 1. As high negative ΔG_bind_ represents great affinity for the receptor, Comp1, Comp2, and Comp3 with binding free energies −64.28, −58.22 and −75.241 kcal/mol, respectively, were found to possess a strong affinity for NS2B-NS3, suggesting its high potential to be an efficient inhibitor of the ZIKV NS2B-NS3 protein.

### 3.6. Molecular Dynamics Simulation

The stability and dynamic behaviour of protein–ligand complexes, was determined by Molecular Dynamics Simulation (MD) studies. High binding free energies-exhibiting top hits, selected based on MM-GBSA analysis, were subjected to MD simulations. These top hits included Comp1, Comp2, and Comp3 with binding free energies of −64.28 kcal/mol, −55.15 kcal/mol and −50.16 kcal/mol, respectively. NS2B-NS3 in complex with each hit was employed, and the projected changes in their conformation from the initial structure over the simulation period were expressed as Root Mean Square Deviation (RMSD). Moreover, Root Mean Square Fluctuation (RMSF) values were also calculated to analyse the structural stability, atomic mobility, and flexibility of residues during protein–hit interaction.

MD simulation studies indicated the protein backbone was found to be consistent during the simulation, and no abnormal deviation was found in comparison to the native structure, which confirmed the binding stability of Comp1 with NS2B-NS3. More precisely, initially the backbone RMSD was consistent until 83 ns; after that, there was a small flip, and then consistency was achieved until the end of simulation (Figure 5). For Comp2-NS2B-NS3 complex, the system was equilibrated, and the backbone was stable until the end of simulation (Figure 6). For Comp3-protein complex, the backbone was also consistent and stable throughout the simulation (Figure 7a). In the case of the ligand (Comp1), there was not much deviation observed, except during about 80 to 85 ns. Higher deviation during the above duration might be due to high level of conformational changes, and after about 85 ns, again, the ligand RMSD was equilibrated (Figure 5). For Comp2 and Comp3, there was not much deviation (Figure 6 and Figure 7a). Hence, the above observation explained the protein–ligand complex stability in the dynamic states, as a positive control co-crystal structure of NS2B-NS3 with boronate (PDB ID:5LC0) was also analysed. The RMSD showed the stability of the complex, as there was a small deviation in the start and then the simulation was equilibrated (Figure 7b).

Moreover, to examine individual amino acid residue fluctuation, the RMSF values for all the complexes were estimated. The RMSF values for the catalytic region of the NS2B-NS3 of all the complexes exhibited stability. There were no considerable fluctuations observed where the ligand binds with the protein. As compared to all residues, the residues Lys 1015, Gly 1016, and Glu 1017 expressed higher fluctuations. The calculated RMSF values are represented in Figure 8, Figure 9, Figure 10 and Figure 11. The green vertical lines in the plots represent the interacting residues of NS2B-NS3 of Zika virus with their respective compounds. The residues where most of the compounds make interactions are SER 81- PHE 84, GLN 135-GLY 139, TRP 150-LYS 154, VAL 172-ASP 175, LEU 1128-THR 1134, and TYR 1150-TYR 1161.

## 4. Discussion

The ZIKV, mosquito-borne flavivirus, belongs to the Flaviviridae family, exhibiting significant similarities to West Nile virus and DENV, as well as the yellow fever virus. Effective and reliable medication or treatment against ZIKV still needs to be discovered. NS2B-NS3 protease is a potential candidate to target the infection caused by the ZIKV because it holds a promising contribution to the viral life cycle. A few studies have reported small molecules, such as flavonoids, bromocriptine, and other small molecules, as inhibitors to target NS2B-NS3 protease, using different virtual drug discovery approaches [17,48,49,50]. There still is a need to discover novel inhibitors for the NS2B-NS3 protein to achieve efficacious treatment for ZIKV infection. The present in silico study aimed at employing e-pharmacophore-based high-throughput virtual screening to discover potent inhibitors targeting ZIKV’s NS2B-NS3 protein. To serve this purpose, the ASINEX database containing screening libraries of lead-like molecules was exploited to retrieve the potential NS2B-NS3 protein inhibitors. The five-featured hypothesis (ADPRR) was generated to screen the database. Furthermore, HTVS, SP, and XP docking were carried out to screen the obtained hit molecules, resulting in six hits with higher Glide scores than boronate inhibitors. Subsequently, the pharmacokinetic parameters were also assessed, including QPlogPo/w, QPlogS, QPPCaco, QPlogBB, QPPMDCK, HOA%, and Lipinski’s rule of five. The findings revealed three hits, i.e., Comp1, Comp2 and Comp3, exhibiting favourable ADME properties. The findings were in accordance with the previous explorations in the search for potential NS2B-NS3 targeting inhibitors [17,37]. Additionally, to validate the docking results, molecular dynamics simulation studies were performed. The findings of the MD results evinced the stability of the NS2B-NS3-hit molecule complexes. The structural analysis detected the presence of an imidazole ring in the screened hits. An imidazole ring and its derivatives have been reported to exhibit effective antiviral activities against several viral diseases, such as dengue virus (DENV), hepatitis C virus (HCV), West Nile virus (WNV), and many others [51,52]. The findings of the present study, in accordance with the previous research, deduced that these hits hold the promising prospect to be candidate lead compounds against the NS2B-NS3 protease of ZIKV.

## 5. Conclusions

The Zika virus outbreak created a global health concern, resulting in the declaration of a pandemic by the World Health Organization in 2016 [53]. The drug discovery studies concerning ZIKV have focused on the identification of certain inhibitors to target the Zika virus to treat its infection. Great emphasis has been imposed to target and inhibit proteases, more typically, the non-structural NS2B-NS3 protein, owing to its significant role in the viral life cycle. Undoubtedly, immense consideration is needed to discover potent and reliable drugs by exploring potential inhibitors of ZIKV NS2B-NS3 protein. The present study has identified the potential NS2B-NS3 protein inhibitors by coupling high throughput screening with MM-GBSA and molecular dynamics simulation. The docking studies showed the docking scores of compound 1, compound 2 and compound 3 are −7.399, −8.040 and −5.792, respectively. The binding energies validated the strong binding as compound 1, compound 2, and compound 3 showed binding free energies of −64.28 kcal/mol, −55.15 kcal/mol and −50.16 kcal/mol, respectively. Furthermore, the RMSD graph also indicated that the protein–ligand complexes were stable during 100 ns of MD simulation. The pharmacokinetic analysis revealed that top-scored inhibitors have no violation of Lipinski’s rule of five, confirming drug-likeness ability. Hence, both docking and molecular dynamics simulation investigations revealed that the top-scored inhibitors have a strong binding affinity towards the NS2B-NS3 protein of ZIKV. The e-pharmacophore-based virtual screening approach resulted in the identification of novel inhibitor candidates of the NS2B-NS3 Zika protein that can be further explored to achieve the effective treatment of ZIKV infection and to combat the rapid spread of Zika virus. Consequently, this study can provide a way for researchers to test these inhibitors against Zika virus in in vitro and in vivo analysis.

## Figures and Tables

**Figure 1 vaccines-11-00131-f001:**
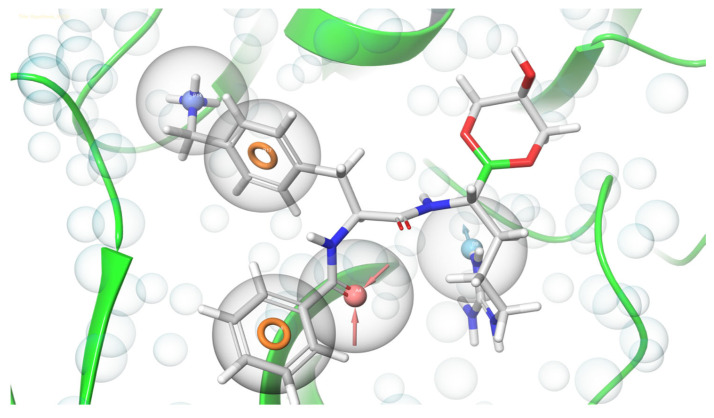
The e-pharmacophore model of NS2B-NS3 protein of ZIKV with boronate inhibitor showing one hydrogen bond acceptor (A), one hydrogen bond donor (D), one positively ionic (P) and two aromatic rings structures (R).

**Figure 2 vaccines-11-00131-f002:**
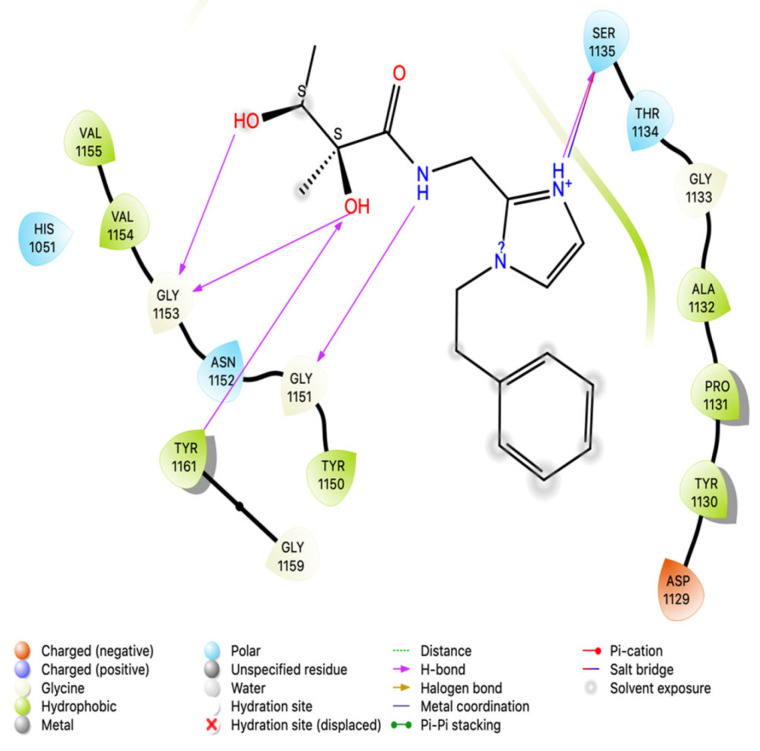
Interactions analysis of comp 1 with NS2B-NS3 protein of Zika virus.

**Figure 3 vaccines-11-00131-f003:**
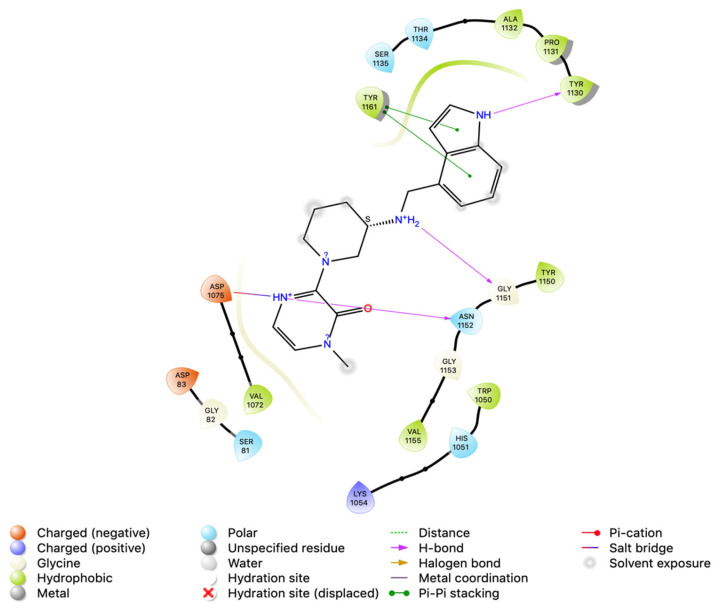
Interactions analysis of comp 2 with NS2B-NS3 protein of Zika virus.

**Figure 4 vaccines-11-00131-f004:**
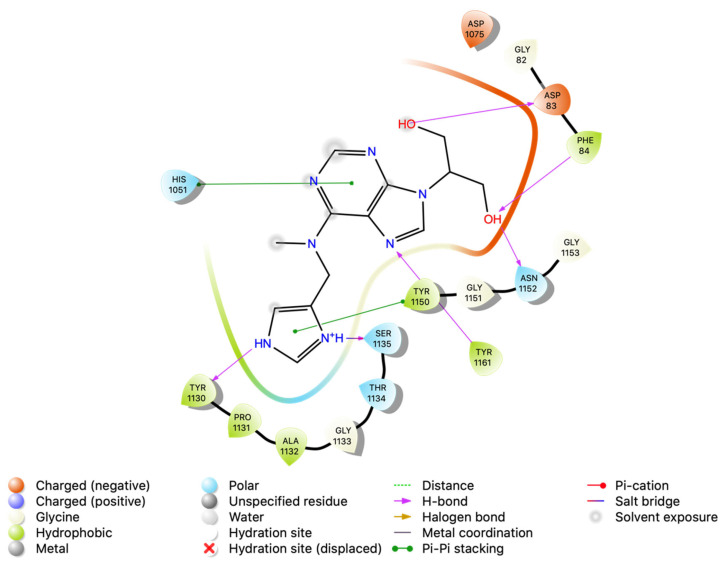
Interactions analysis of comp 3 with NS2B-NS3 protein of Zika virus.

**Figure 5 vaccines-11-00131-f005:**
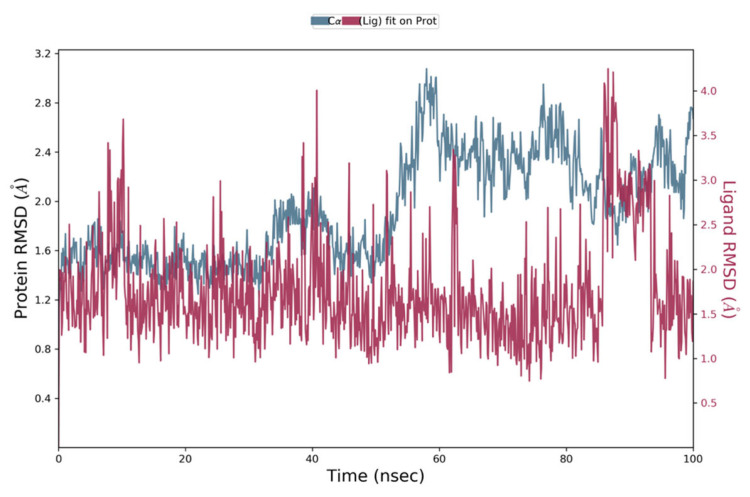
Root Mean Square Deviation plot of Comp1 and NS2B-NS3 complex of ZIKV.

**Figure 6 vaccines-11-00131-f006:**
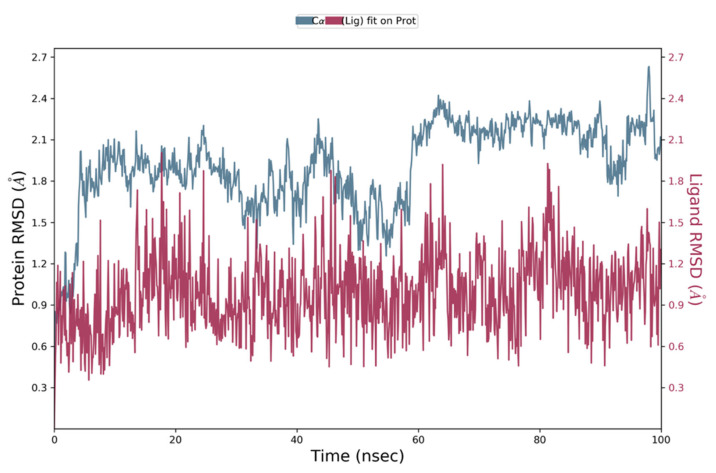
Representation of Root Mean Square Deviation plot of Comp2 and NS2B-NS3 complex of ZIKV.

**Figure 7 vaccines-11-00131-f007:**
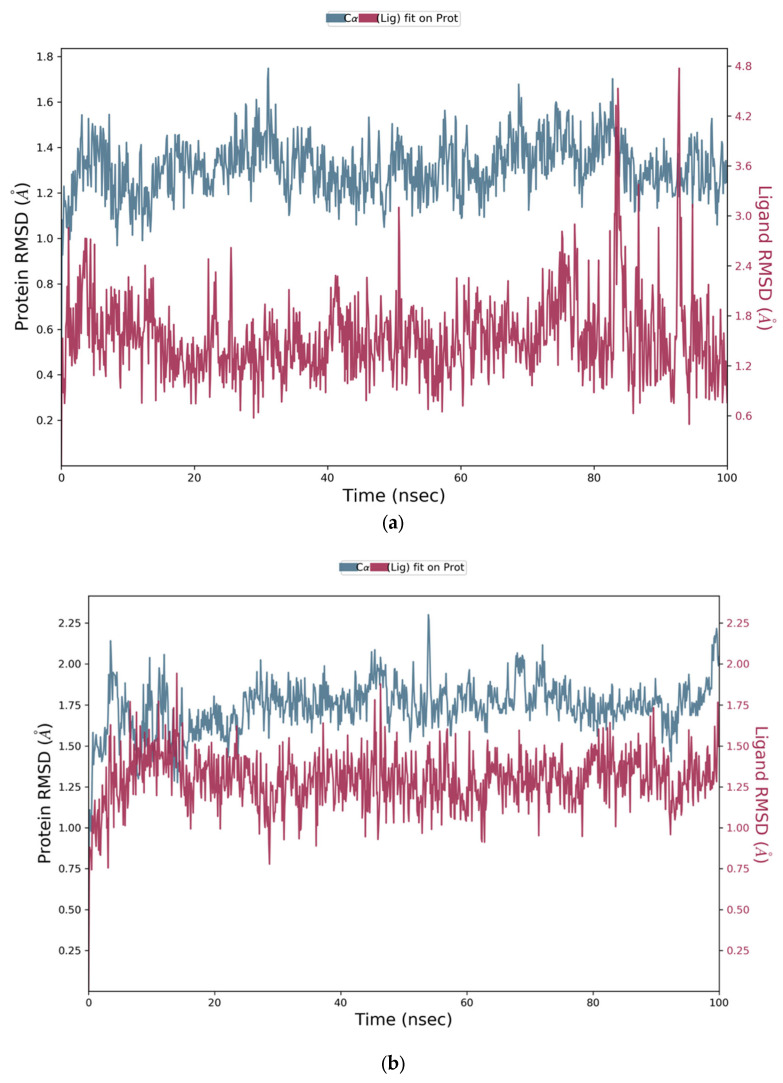
(**a**) Representation of Root Mean Square Deviation plot of Comp3 and NS2B-NS3 complex of ZIKV. (**b**) Root Mean Square Deviation plot of NS2B-NS3 complex of ZIKV PDB ID: 5LC0.

**Figure 8 vaccines-11-00131-f008:**
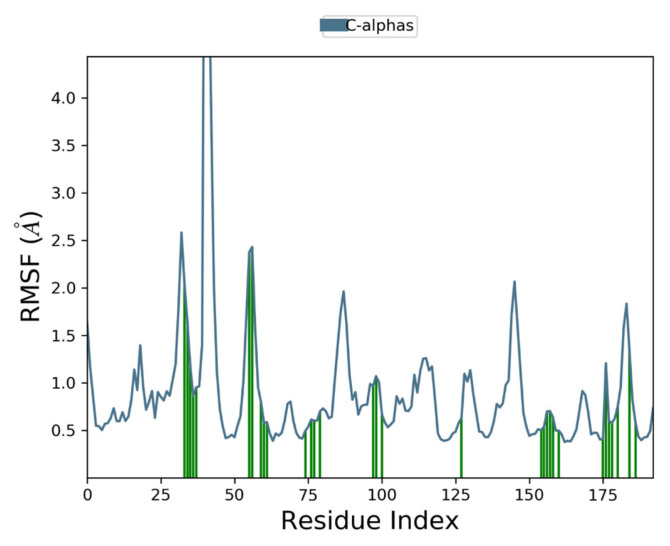
Root Mean Square Fluctuation plot of Comp1 and NS2B-NS3 complex of ZIKV.

**Figure 9 vaccines-11-00131-f009:**
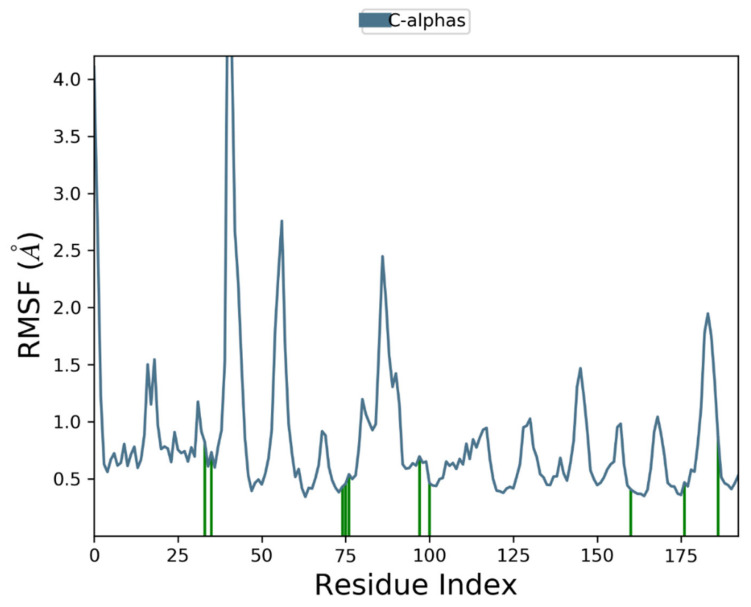
Root Mean Square Fluctuation plot of Comp2 and NS2B-NS3 complex of ZIKV.

**Figure 10 vaccines-11-00131-f010:**
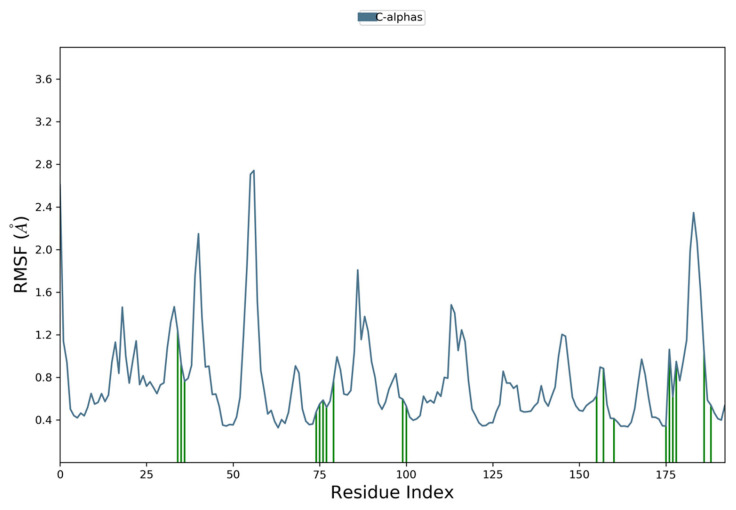
Root Mean Square Fluctuation plot of Comp3 and NS2B-NS3 complex of ZIKV.

**Figure 11 vaccines-11-00131-f011:**
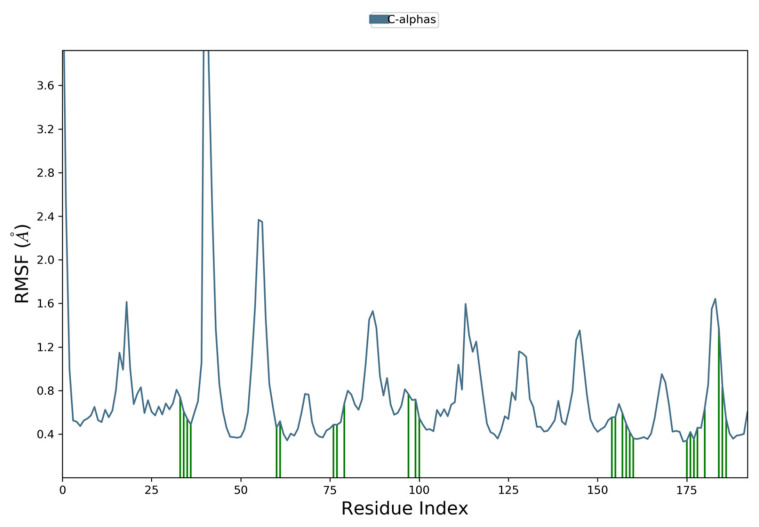
Root Mean Square Fluctuation plot NS2B-NS3 complex of ZIKV PDB ID: 5LC0.

**Table 1 vaccines-11-00131-t001:** Top 3 hits with their Glide score and MMGBSA values.

Compound	XP GScore	MMGBSA	Mol MW	Docking Score	Rule of Five Violation	Prime vdw
Comp 1	−7.399	−64.58	317.387	−7.228	0	−833.91
Comp 2	−8.040	−55.15	337.424	−8.040	0	−828.84
Comp 3	−5.792	−50.16	303.323	−5.792	0	−829.43

**Table 2 vaccines-11-00131-t002:** ADME/T properties of all three compounds select after HTVS.

Ranges Sr. No.	QPlogPo/w (−2.0–6.5)	QPlogBB (−3.0–1.2)	Mol MW (130.0–725)	Percent Human-Oral Absorption (>80% Is High <25% Is Poor)	FISA (7.0–330)	PISA (0.0–450)	QPlogHERG (Concern below −5)	QPlogS (−6.5–0.5)
Comp 1	2.307	0.158	342.827	90.439	115.470	242.988	−3.454	−2.855
Comp 2	0.681	−2.206	343.359	58.803	257.159	304.534	−5.85	−3.726
Comp 3	0.288	−1.553	303.323	69.379	181.286	166.144	−4.592	−2.086

## Data Availability

More data related to this study can be accessed by contacting the corresponding author.

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
