# Peer review of "Identification of NS2B-NS3 Protease Inhibitors for Therapeutic Application in ZIKV Infection: A Pharmacophore-Based High-Throughput Virtual Screening and MD Simulations Approaches"

_vaccines, 2023, doi:10.3390/vaccines11010131_

Round 1

Reviewer 1 Report

This manuscript can be accepted after major revision.

Here are the points:

·       Authors should write the long version of ADME in first place that it is mentioned.

·       Authors should enlighten why they created ADPRR from the large possibilities in Phase.

·       It is not clear why they retrieved the crystal structure of the NS2B-NS3 protein with PDB ID: 5LC0. Did they try with other NS2B-NS3 protein with different codes?

·       It is not clear if they made re-docking with the ligand of protein (PDB ID: 5LC0) or not.

·       Authors should add specified ranges of each parameter of ADME in Table 2. Authors should put the results of Lipinski’ Rule of 5.

·       Authors should also discuss why they choose these ADME properties what is the importance for ZIKV infection.

Author Response

We are thankful to the worthy reviewer and highly appreciate for improving our Manuscript. The questions and concerns raised by the worthy reviewers are now addressed to our best

Reviewer 1:

Reviewers’ Comments

Changes/Comments

 Authors should write the long version of ADME in first place that it is mentioned.

Incorporated

Authors should enlighten why they created ADPRR from the large possibilities in Phase.

Energy-optimised pharmacophore (E-Pharmacophore) is an automated module in Phase of Schrodinger suite. An energy-optimized pharmacophore hypothesis was generated based on the protein-ligand complex. The hypothesis assisted to derive electrostatic and steric features of the ligands with binding sites. So, by scanning protein-ligand complex and by considering the electrostatic and steric features of the ligands with binding sites, an automated e-pharmacophore has been generated by Phase.

It is not clear why they retrieved the crystal structure of the NS2B-NS3 protein with PDB ID: 5LC0. Did they try with other NS2B-NS3 protein with different codes?

It is co-crystalized structure with inhibitor and most of the research article used this structure for Virtual Screening campaign. The reference has been cited in the paper accordingly. Following is the one of the recent articles.

Mirza, M. U., Alanko, I., Vanmeert, M., Muzzarelli, K. M., Salo-Ahen, O. M., Abdullah, I., ... & Froeyen, M. (2022). The discovery of Zika virus NS2B-NS3 inhibitors with antiviral activity via an integrated virtual screening approach. European Journal of Pharmaceutical Sciences, 106220

 It is not clear if they made re-docking with the ligand of protein (PDB ID: 5LC0) or not.

Yes, we have made redocking and updated the Manuscript accordingly

Authors should add specified ranges of each parameter of ADME in Table 2. Authors should put the results of Lipinski’ Rule of 5.

       Authors should also discuss why they choose these ADME properties what is the importance for ZIKV infection.

Incorporated accordingly

Reviewer 2 Report

The manuscript ID vaccines-2105490 entitled " Identification of NS2B-NS3 Protease Inhibitors for Therapeutic application in ZIKV infection: A Pharmacophore-based High-throughput Virtual Screening and MD Simulations Approaches" is a good study. Here, in-silico exploration aimed to identify the novel hit to inhibit the Zika NS2B-NS3 protease using e-pharmacophore-based high-throughput virtual screening. A 3D e-pharmacophore model is generated based on the five-featured (ADPRR) pharmacophore hypothesis. Subsequently, the predicted model is further subjected to high-throughput virtual screening to obtain top-hit molecules from the chemical databases. Initial hits were examined in terms of binding free energies and ADME properties to identify the candidate hit exhibiting a favorable pharmacokinetic profile. Eventually, molecular dynamic (MD) simulation studies are conducted to examine the stability of the hit molecule inside the receptor cavity. The findings of the in-silico analysis manifested affirmative evidence for the hit molecules with -64.28, -55.15, and − 50.16 kcal/mol binding free energies, as potent inhibitors for the Zika NS2B-NS3 protease. Hence, it holds the striking potential to serve as a prospective inhibitor for designing efficacious drugs against ZIKV and related viral infections.

I appreciate the authors' effort to study ZIKV-NS2B-NS3 Protease Inhibitors. However, the following comments need to be addressed

1)      Abstract: Unclear on the results section. The conclusion is not convincing.

2)      The methodology is poor; How did the authors' selected protein structure? 5LC0? what is the IC50 value of the inbound inhibitors?

3)      Describe in detail- the ASINEX database.

4)      Order is misplaced:  2.4. Molecular Docking and 2.5. Energy-optimized pharmacophore----it may be reversed as per abstract????? is its structure or ligand-based pharmacophore? how efficient is the e-pharmacophore?

5)      The methodology for Molecular Dynamics Simulation is poor. How was the complex prepared for the MD?

6)      Did the results compare with any active inhibitors? it may be meaningful.

7)      Figure 1-legends is missing. Include all the elements in the figure.

8)      What are the key residues observed in Figures 8,9,10 &11?

9)      All the results should be discussed properly.

10)   The conclusion needs revision according to the results of the present study and future perspectives.

Author Response

We are thankful to the worthy reviewer and highly appreciate for improving our Manuscript. The questions and concerns raised by the worthy reviewers are now addressed to our best

Reviewer 2:

Reviewers’ Comments

Changes/Comments

Abstract: Unclear on the results section. The conclusion is not convincing

Updated accordingly

 The methodology is poor; How did the authors' selected protein structure? 5LC0? what is the IC50 value of the inbound inhibitors?

It is co-crystalized structure with inhibitor and most of the research article used this structure for Virtual Screening campaign. The reference has been cited in the paper accordingly. Following is the one of the recent articles.

Mirza, M. U., Alanko, I., Vanmeert, M., Muzzarelli, K. M., Salo-Ahen, O. M., Abdullah, I., ... & Froeyen, M. (2022). The discovery of Zika virus NS2B-NS3 inhibitors with antiviral activity via an integrated virtual screening approach. European Journal of Pharmaceutical Sciences, 106220

IC 50 and Ki values has been incorporated in the Manuscript accordingly

Describe in detail- the ASINEX database

Updated accordingly

Order is misplaced:  2.4. Molecular Docking and 2.5. Energy-optimized pharmacophore----it may be reversed as per abstract????? is its structure or ligand-based pharmacophore? how efficient is the e-pharmacophore?

First the inbound ligand was redocked with the protein then an energy-optimized pharmacophore (E-pharmacophore) hypothesis was generated based on the protein-ligand complex. Energy-optimised pharmacophore is an automated module in Phase of Schrodinger suite.  The e-Pharmacophores method achieves the advantages of both ligand- and structure-based approaches by generating energetically optimized, structure-based pharmacophores that can be used to rapidly screen millions of compounds.

The methodology for Molecular Dynamics Simulation is poor. How was the complex prepared for the MD?

Updated accordingly

Did the results compare with any active inhibitors? it may be meaningful

The boronate inhibitor has been used as control

Figure 1-legends is missing. Include all the elements in the figure

Updated accordingly

What are the key residues observed in Figures 8,9,10 &11?

Updated accordingly

 All the results should be discussed properly

Updated accordingly

 The conclusion needs revision according to the results of the present study and future perspectives.

The conclusion has been revised and updated accordingly

Round 2

Reviewer 1 Report

The authors have answered properly to all points and in my opinion the paper can now be accepted.